

# Fine particulate matter (PM$_{2.5}$) trends in China, 2013-2018: contributions from meteorology

Shixian Zhai[1,2], Daniel J. Jacob[2], Xuan Wang[2], Lu Shen[2], Ke Li[2], Yuzhong Zhang[2], Ke Gui[3], Tianliang Zhao[1], Hong Liao[4]

[1]Key Laboratory for Aerosol-Cloud-Precipitation of China Meteorological Administration, Collaborative Innovation Center on Forecast and Evaluation of Meteorological Disasters, School of Atmospheric Physics, Nanjing University of Information Science & Technology, Nanjing 210044, China

[2]John A. Paulson School of Engineering and Applied Sciences, Harvard University, Cambridge, Massachusetts 02138, USA

[3]Key Laboratory for Atmospheric Chemistry, Chinese Academy of Meteorological Sciences, CMA, Beijing 100081, China

[4]Jiangsu Key Laboratory of Atmospheric Environment Monitoring and Pollution Control, Collaborative Innovation Center of Atmospheric Environment and Equipment Technology, School of Environmental Science and Engineering, Nanjing University of Information Science & Technology, Nanjing 210044, China

*Correspondence to*: Shixian Zhai (zhaisx@g.harvard.edu).

**Abstract.** Fine particulate matter (PM$_{2.5}$) is a severe air pollution problem in China. Observations of PM$_{2.5}$ have been

available since 2013 from a large network operated by the China National Environmental Monitoring Center (CNEMC). The

data show a general 30-50% decrease of annual mean PM$_{2.5}$ across China over the 2013-2018 period, averaging 5.2 μg m$^{-3}$ a$^{-1}$. Trends in the five megacity cluster regions targeted by the government for air quality control are $9.3 \pm 1.8$ μg m$^{-3}$ a$^{-1}$ ($\pm$

95% confidence interval) for Beijing-Tianjin-Hebei, $6.1 \pm 1.1$ μg m$^{-3}$ a$^{-1}$ for Yangtze River Delta, $2.7 \pm 0.8$ μg m$^{-3}$ a$^{-1}$ for

Pearl River Delta, $6.7 \pm 1.3$ μg m$^{-3}$ a$^{-1}$ for Sichuan Basin, and $6.5 \pm 2.5$ μg m$^{-3}$ a$^{-1}$ for Fenwei Plain (Xi'an). Concurrent 2013-

2018 observations of sulfur dioxide (SO$_2$) and CO show that the declines in PM$_{2.5}$ are qualitatively consistent with drastic

controls of emissions from coal combustion. However, there is also a large meteorologically driven interannual variability of

PM$_{2.5}$ that complicates trend attribution. We used a stepwise multiple linear regression (MLR) model to quantify this

meteorological contribution to the PM$_{2.5}$ trends across China. The MLR model correlates the 10-day PM$_{2.5}$ anomalies to wind

speed, precipitation, relative humidity, temperature, and 850 hPa meridional wind velocity (V850). We find that meteorology

made a minor but significant contribution to the observed 2013-2018 PM$_{2.5}$ trends across China and that removing this

influence reduces the uncertainty on the emission-driven trends. The mean PM$_{2.5}$ decrease across China is 4.6 ug m$^{-3}$ a$^{-1}$ in

the meteorology-corrected data, 12% weaker than in the original data. The residual trends in the five megacity clusters

attributable to changes in anthropogenic emission are $8.0 \pm 1.1$ μg m$^{-3}$ a$^{-1}$ for Beijing-Tianjin-Hebei (14% weaker than the

observed trend), $6.3 \pm 0.9$ μg m$^{-3}$ a$^{-1}$ for Yangtze River Delta (3% stronger), $2.2 \pm 0.5$ μg m$^{-3}$ a$^{-1}$ for Pearl River Delta (19%





weaker), $4.9 \pm 0.9$ µg m$^{-3}$ a$^{-1}$ for Sichuan Basin (27% weaker), and $4.9 \pm 1.9$ µg m$^{-3}$ a$^{-1}$ for Fenwei Plain (Xi'an; 25%

weaker). 2015-2017 observations of flattening PM$_{2.5}$ in the Pearl River Delta, and increase in the Fenwei Plain, can be

attributed to meteorology rather than to relaxation of emission controls.

## 1.    Introduction

PM$_{2.5}$ (particulate matter with aerodynamic diameter less than 2.5µm) is a severe air pollution problem in China, responsible

for 1.1 million excess deaths in 2015 (Cohen et al., 2017). The Chinese government introduced in 2013 the Action Plan on

the Prevention and Control of Air Pollution (Chinese State Council, 2013a), called Clean Air Action for short, to

aggressively control anthropogenic emissions. Starting that year, PM$_{2.5}$ data from a nationwide monitoring network of about

1,000 sites also became available from the China National Environmental Monitoring Center (CNEMC) of the Ministry of

Ecology and Environment of China (MEEC). These data show 30-40% decreases of PM$_{2.5}$ across eastern China over the

2013-2017 period (Chinese State Council, 2018a; X. Zhang et al., 2019). However, interpretation of these trends in terms of

emission controls may be complicated by interannual variability and trends in meteorology (R. Zhang et al., 2014; Y. Wang

et al., 2014; Zhu et al., 2012; Jia et al., 2015; K. Li et al., 2018; Yang et al., 2018; Yang et al., 2016; Cheng et al., 2018; Chen

et al., 2019). Here we use a stepwise multi-linear regression (MLR) model to separate the effects of meteorological

variability and emission controls on the 2013-2018 trends in PM$_{2.5}$ across China.

Meteorology drives large day-to-day, seasonal, and interannual variations in PM$_{2.5}$ in China by affecting transport,

scavenging, emissions, and chemical production (Y. Wang et al., 2014; Leung et al., 2018; Tai et al., 2012). The relationships

between PM$_{2.5}$ and meteorological variables are complex and differ by region and time of year (Shen et al., 2017). For

example, wintertime PM$_{2.5}$ pollution events in central and eastern China are associated with low wind speed and high relative

humidity (RH) (Wang et al., 2014; R. Zhang et al., 2014; Pendergrass et al., 2019; Moch et al., 2018; Song et al., 2019). On

the other hand, high wind speeds in northern China in spring and summer promote dust emission (Lyu et al., 2017; X. Wang

et al., 2004). Precipitation scavenging is a major factor driving PM$_{2.5}$ variability in southern and coastal China (Chen et al.,

2018; Leung et al., 2018). The 850-hPa meridional wind velocity (V850) is strongly correlated with PM$_{2.5}$ in the North China

Plain (Pendergrass et al., 2019; Shen et al., 2018).

Anthropogenic emissions of PM$_{2.5}$ and its precursors including sulfur dioxide (SO$_2$), nitrogen oxides (NO$_x$), ammonia (NH$_3$),

and nonmethane volatile organic compounds (NMVOCs) have undergone large changes in China over the past decades.

Rapid growth in emissions from 1980 to 2006 led to a general increase in PM$_{2.5}$ over China, as demonstrated by visibility

data (Che et al., 2007; Han et al., 2016; Wang and Chen, 2016; Fu et al., 2014; Zhang et al., 2012) and since 1999 by satellite





aerosol optical depth (AOD) data (Ma et al., 2016; Lin et al., 2018; Zhao et al., 2017). $SO_2$ emissions peaked in 2006/2007,

$NO_x$ emissions peaked in 2011, and $NH_3$ emissions peaked around 1996, as estimated from emission inventories (Zhao et al.,

2017; J. Wang et al., 2017b; Xia et al., 2016; Liu et al., 2016a; Lu et al., 2010; Xu et al., 2016; Kang et al., 2016) and

observed from satellites (Xia et al., 2016; F. Liu et al., 2016a; de Foy et al., 2016; van der A et al., 2017). $SO_2$ and $NO_x$

emissions have declined since their peaks, whereas $NH_3$ emissions have remained relatively stable (Zhao et al., 2017). The

onset of emission controls led to slight decreases in $PM_{2.5}$ over the 2006-2012 period as indicated by satellite AOD data (Ma

et al., 2016; Lin et al., 2018; Zhao et al., 2017) and surface observations (Tao et al., 2017; Wang et al., 2017). The Clean Air

Action greatly increased the scope of emission controls. The Multi-resolution Emission Inventory for China (Zheng et al.,

2018) (MEIC, http://www.meicmodel.org) estimates nationwide decreases over the 2013-2017 period of 59% for $SO_2$, 33%

for $PM_{2.5}$, 21% for $NO_x$, and 3% for $NH_3$, with NMVOCs increasing by 2%. Continued reductions in emissions are required

and implemented in 2018 (Chinese State Council, 2018b). Quantifying the response of $PM_{2.5}$ to these rapid emission changes

by resolving the effect of meteorological variability is an important question for measuring the success of the Clean Air

Action.

## 2.    Data and methods

### 2.1.    Observations

We use 2013-2018 hourly data for surface air $PM_{2.5}$ together with $SO_2$, nitrogen dioxide ($NO_2$), and carbon monoxide (CO)

concentrations from the CNEMC network (http://106.37.208.233:20035/). The network started in January 2013 with 496 sites

in 74 major cities across the country (Chinese State Council, 2013b), growing to 1497 sites in 454 cities by 2018. $PM_{2.5}$ mass

concentrations are measured using the micro oscillating balance method and/or the β absorption method (MEP, 2012; Zhang

and Cao, 2015). $SO_2$, $NO_2$, and CO concentrations are measured at the same sites as $PM_{2.5}$. $NO_2$ concentrations are measured

by the molybdenum converter method known to have positive interferences from $NO_2$ oxidation products (Dunlea et al., 2007).

$SO_2$ and CO are respectively measured using ultraviolet fluorescence and infrared absorption (MEP, 2012; Zhang and Cao,

2015). We applied quality control to the hourly CNEMC data following Barrero et al. (2015) to exclude severe outliers. This

removed 6.7%, 5.7%, 5.7%, and 5.9% of the $PM_{2.5}$, $SO_2$, $NO_2$, and CO data respectively.

We correlated these air quality observations with meteorological observations from 839 stations distributed across China

(Figure S1) and compiled in the Surface Daily Climate Dataset (V3.0) released by the China National Meteorological

Information Center (CNMIC; http://data.cma.cn/). These include data for wind speed (WDS), precipitation (PRECIP), relative

humidity (RH), and temperature (TEM). We also used the 850-hPa meridional wind velocity (V850) from the MERRA-2





reanalysis produced at 0.5ºx0.625º horizontal resolution by the NASA Global Modeling and Assimilation Office (https://gmao.gsfc.nasa.gov/reanalysis/MERRA-2). We choose these meteorological variables for their strong correlations with $PM_{2.5}$ identified in previous studies.

All data in this work are averaged over 10 days (10-day time resolution). Trend analyses use only those sites with at least

70% data coverage for each year of the 2013-2018 period. For the MLR model, we further average all data on a 2º×2.5º grid to increase statistical robustness (Tai et al., 2012).

The 2013 Clean Air Action (Council, 2013a) identified three megacity clusters as target regions for reducing air pollution: Beijing-Tianjin-Hebei (BTH; 35-41°N, 113.75-118.75°E), Yangtze River Delta (YRD; 29-33°N, 118.75-123°E), and Pearl River Delta (PRD; 21-25°N, 111.25-116.25°E). The more recent plan released in July 2018 (Chinese State Council, 2018b)

removed PRD from the list of target regions and added Fenwei Plain (FWP; 33-35°N, 106.25-111.25°E & 35-37°N, 108.75-113.75°E). Previous studies (X. Zhang et al., 2012) also identified Sichuan Basin (SCB; 27-33°N, 103.75-108.75°E) as one of the major haze regions in China. We present analyses for these five target regions by averaging the data from all sites with more than 70% data coverage for each year of 2013-2018. The only continuous record for 2013-2018 in the FWP region is for Xi'an (13 sites). Additional FWP sites outside Xi'an started operating in early 2015 and are consistent with the Xi'an

data, as will be shown below.

## 2.2.    Multiple linear regression model

We construct a stepwise multiple linear regression (MLR) model to quantify the effect of meteorology on $PM_{2.5}$ variability. The model fits the deseasonalized and detrended 10-day $PM_{2.5}$ mean time series on the 2º×2.5º grid to the five deseasonalized and detrended 10-day mean meteorological variables (WDS, PRECIP, RH, TEM, and V850). The deseasonalized and

detrended time series are obtained by subtracting the 50-day moving averages from the 10-day mean time series (Tai et al., 2010). This focuses on synoptic scales of variability and avoids aliasing from common seasonal variations and long-term trends between variables (Shen et al., 2017).

Separate fits of $PM_{2.5}$ to the meteorological variables are done for each 2º×2.5º grid square and season (DJF, MAM, JJA, SON). The fit has the form:


$$Y_{d,i}(t) = \sum_{k=1}^{5} \beta_{i,k} X_{d,i,k}(t) + b_i \qquad (1)$$

where $Y_{d,i}(t)$ is the deseasonalized and detrended $PM_{2.5}$ time series for grid square and season $i$, and $X_{d,i,k}(t)$ is the corresponding time series for the deseasonalized and detrended meteorological variable $k \in [1,5]$. We fit the regression



coefficients $\beta_{i,k}$ and the intercept $b_i$. The regression is done stepwise to add and delete terms based on their independent

statistical significance to obtain the best model fit (Drapper and Smith, 1998). The fits and the selected meteorological variables

differ by location and season but with regional consistency (Table S1). For meteorological variables not in the final MLR

model, the regression coefficients $\beta_{i,k}$ in equation (1) are zero.

### 2.3.    Application to 2013-2018 PM$_{2.5}$ trends

We use the MLR model to remove the effect of meteorological variability from the 2013-2018 PM$_{2.5}$ trends, including not only

the 10-day synoptic-scale variability but also any interannual variability and 6-year trends. This makes the standard assumption

that the same factors that drive synoptic-scale variability also drive interannual variability (Jacob and Winner, 2009; Tai et al.,

2012). We thus apply equation (1) to the meteorological anomalies $X_{a,i,k}$, obtained by removing the 6-year means of the 50-day

moving average. The anomalies are deseasonalized but not detrended. This yields the meteorology-driven PM$_{2.5}$ anomalies

$Y_{m,i}$:

$$Y_{m,i}(t) = \sum_{k=1}^{5} \beta_{i,k} X_{a,i,k}(t) + b_i \tag{2}$$

Consider now the PM$_{2.5}$ anomaly $Y_{a,i}$ for grid square and season $i$ obtained by deseasonalizing but not detrending the PM$_{2.5}$

data, in the same way as for the meteorological variables. The residual anomaly $Y_{r,i}$ after removing meteorological influence

from the MLR model is given by

$$Y_{r,i}(t) = Y_{a,i}(t) - Y_{m,i}(t) \tag{3}$$

The residual is the component of the anomaly that cannot be explained by the MLR meteorological model. It includes noise

due to limitations of the MLR model and other factors, but also a long-term trend over the 6-year period that we attribute to

changes in anthropogenic emissions. The same approach was recently applied by K. Li et al. (2019) to separate anthropogenic

and meteorological drivers of ozone trends in China.

### 3.    Results and Discussion

### 3.1.    PM$_{2.5}$ trends in China, 2013-2018

Figure 1 shows annual mean observed PM$_{2.5}$ concentrations from the CNEMC over China for 2013 and 2018, and the linear

regression trends on the $2°\times2.5°$ grid based on the PM$_{2.5}$ anomalies $Y_{a,i}(t)$ including effects of both changing emissions and

meteorology. In 2013, PM$_{2.5}$ across most of China was well above the Chinese national air quality standard (annual mean of

35 µg m$^{-3}$). BTH and FWP (Xi'an) had the highest PM$_{2.5}$ among the five target regions, with annual average concentrations of



$108 \pm 34$ μg m$^{-3}$ (standard deviation describes variability of the annual average across sites in the region) and $108 \pm 11$ μg m$^{-3}$

respectively, followed by SCB ($72 \pm 17$ μg m$^{-3}$), YRD ($67 \pm 12$ μg m$^{-3}$), and PRD ($47 \pm 7$ μg m$^{-3}$). PM$_{2.5}$ decreased dramatically

from 2013 to 2018, by 34-49% for the five target regions. Mean 2018 concentrations were $55 \pm 13$ μg m$^{-3}$ in BTH, $62 \pm 4$ μg

m$^{-3}$ in FWP (Xi'an), $41 \pm 7$ μg m$^{-3}$ in SCB, $41 \pm 15$ μg m$^{-3}$ in YRD, and $31 \pm 5$ μg m$^{-3}$ in PRD.

Figure 2 shows the 2013-2018 relative trends of annual mean PM$_{2.5}$ for the five target regions, along with the corresponding

trends of SO$_2$, NO$_2$, and CO concentrations measured at the same sites. Also shown in the bottom panels are the MEIC

inventory trends in emissions of primary PM$_{2.5}$, SO$_2$, NO$_x$, NH$_3$, and CO for 2013-2017. The PM$_{2.5}$ observations show steady

decreases for BTH, YRD, and SCB. PRD flattens out in 2015-2017 before decreasing again in 2018. FWP (Xi'an) decreases

sharply by 47% from 2013 to 2015 but rebounds in 2015-2017 before decreasing again in 2018. Trends at other FWP sites

that became operational in early 2015 are similar to Xi'an. We argue in Section 3.3 that the 2015-2017 flattening at PRD and

the anomalously 2013-2015 sharp decrease and 2015-2017 rebound at FWP are driven by meteorology.

We see from Figure 2 that only SO$_2$ has a decrease steeper than PM$_{2.5}$, indicating that SO$_2$ emission controls have been a

major driver of the PM$_{2.5}$ trend (Lang et al., 2017; Shao et al., 2018). The overall SO$_2$ decrease for the five regions is 57-76%

from 2013 to 2018. The SO$_2$ decrease is quantitatively consistent with the decrease of SO$_2$ emissions estimated by MEIC

(Zheng et al., 2018). This drastic cut of China SO$_2$ emissions is due to installation of scrubbers at coal-fired power plants

(Siwen et al., 2015; Karplus et al., 2018; Silver et al., 2018), elimination of small coal boilers, improvement of coal quality

(Zheng et al., 2018), and switch from residential coal to clean fuels (Zhao et al., 2018). We also see a significant decrease in

CO of 18-43% for the five regions from 2013 to 2018, again consistent with the MEIC emission inventory and suggesting a

reduction in organic PM$_{2.5}$ emissions. Primary PM$_{2.5}$ emissions in the MEIC inventory decreased at a rate comparable or

steeper than CO.

Figure 3 shows the time series of monthly mean PM$_{2.5}$ for the five target regions, illustrating the seasonal and interannual

variability. All regions show winter maxima that can be mostly attributed to meteorology including shallower mixing depth,

lower precipitation, and increased stagnation in winter (X. Wang et al., 2018). Residential heating emissions in winter also

contribute to the seasonality in northern China (Liu et al., 2016b; Xiao et al., 2015). There is a large interannual variability,

particularly in winter, that must be largely driven by meteorology. Studies for BTH have shown that high PM$_{2.5}$ in winter

months is associated with weak southerly winds, low mixing depths, and high relative humidity (Zhang et al., 2014; Chang

et al., 2016; K. Li et al., 2018; Shao et al., 2018). The relatively clean 2017-2018 winter was due in part to a higher

frequency of northerly flow and associated ventilation (Administration, 2018; Yi et al., 2019). In addition, particularly




aggressive actions by the government to restrict coal use that winter may have played a role in reducing PM$_{2.5}$ levels (X. Zhang et al., 2019).

### 3.2. Meteorological influence on PM$_{2.5}$

Figure 4 shows the correlations of 10-day PM$_{2.5}$ concentrations with the individual meteorological variables used in the MLR model. Wind speed is negatively correlated with PM$_{2.5}$, as would be expected from ventilation, except in areas of the north where wind promotes dust formation (Lyu et al., 2017; X. Wang et al., 2004). Precipitation is also generally negatively correlated with PM$_{2.5}$, as one would expect from scavenging (Chen et al., 2018). The positive correlation between precipitation and PM$_{2.5}$ over north China in spring is likely a result of high RH associated with precipitation in adjacent days.

Correlation between RH and PM$_{2.5}$ is positive over northern China, especially in winter, and negative over southern China, especially in summer. The positive correlation between PM$_{2.5}$ and RH over northern China in winter has been reported by previous studies and attributed in part to the role of aqueous-phase aerosol chemistry in driving secondary PM$_{2.5}$ formation (Zheng et al., 2015; He et al., 2018; Song et al., 2018; Pendergrass et al., 2019; Tie et al., 2017). The negative correlation of PM$_{2.5}$ with RH over south China likely reflects the association of high RH with precipitation and onshore wind, which

facilitate PM$_{2.5}$ wet removal and ventilation (Zhu et al., 2012; Leung et al., 2018).

Temperature has a positive correlation with PM$_{2.5}$ year round over most of China (Y. Wang et al., 2014; Leung et al., 2018), even though there is no strong direct dependence of PM$_{2.5}$ on temperature (Jacob and Winner, 2009). The correlation likely reflects the covariation of temperature with the other meteorological variables (Tai et al., 2012; Zhu et al., 2012). A possible explanation for the negative correlation with temperature in summer over North China Plain could be the volatilization of

ammonium nitrate at high temperature (Kleeman, 2008). V850 shows strong positive correlations with winter PM$_{2.5}$ over most of China, and strong negative correlations with summer PM$_{2.5}$ over south China, especially for the Pearl River Delta.

Figure 5 (left panel) describes the ability of the MLR model to account for PM$_{2.5}$ variability in relation to wind speed, precipitation, RH, temperature, and V850 as potential predictor variables. Results are presented as the coefficients of determination $R^2$ (fraction of variance explained) between observed and model PM$_{2.5}$ in the detrended deseasonalized time

series. The $R^2$ values have been adjusted to account for different numbers of significant explanatory terms (predictor variables). $R^2$ values for the five target regions are: 0.59 (BTH), 0.46 (YRD), 0.65(PRD), 0.65 (SCD), and 0.41 (FWP). The right panel of Figure 5 shows the meteorology-corrected PM$_{2.5}$ trends after removal of meteorological variability predicted by the MLR model, i.e., the trends in the residuals $Y_{r,i}(t)$ in equation (3). The meteorology-corrected decreasing trend





averaged across China is -4.6 μg m$^{-3}$ a$^{-1}$, 12% weaker than the trend in the original data (Figure 1) and nowhere contributing

more than 50% to the trend. We elaborate below for the five target regions.

### 3.3.    Meteorology-corrected PM$_{2.5}$ trends for the five target regions

Figure 6 shows the 10-day mean PM$_{2.5}$ anomalies in the deseasonalized (but not detrended) data for the five target regions

($Y_a(t)$ in Section 2.3). Also shown is the meteorological component $Y_m(t)$ derived from the MLR meteorological model, and

the residual $Y_r(t)$ (meteorology-corrected, equation (3)) whose long-term trend can be interpreted as due to changes in

anthropogenic emissions. The PM$_{2.5}$ anomalies show large features on ten-day basis that can be mostly captured by the MLR

model. The residual meteorology-corrected time series is much smoother, as depicted by the narrower 95% confidence

intervals in the anthropogenic residual trends than in the original observed trends. The meteorology-corrected trends differ

by 3% (YRD) to 27% (SCB) from the observed trends. The YRD trend reflects a significant contribution from the December

2013 outlier, which reflects unfavorable meteorological conditions (Figure S2) that are not adequately captured by the MLR

model. If we exclude this outlier month from the time series, the observed YRD trend becomes $5.7 \pm 0.9$ μg m$^{-3}$ a$^{-1}$ and the

meteorology-corrected trend becomes $5.9 \pm 0.7$ μg m$^{-3}$ a$^{-1}$.

Most remarkably, it appears that the 2015-2017 flattening in the PRD and 2015-2017 increase in the FWP can be mostly

attributed to meteorological variability as resolved by the MLR model, rather than to emissions. The trend in the residual is

more consistent with a steady 2013-2018 anthropogenic decrease in both regions. The MLR model shows that meteorology

accelerated the PM$_{2.5}$ decline in PRD and FWP from 2013 to 2015, and contributed partly to the 2015-2017 PM$_{2.5}$ rebound

over FWP. In particular, the high PM$_{2.5}$ anomalies in PRD in 2013 and early 2014 are driven by anomalously low V850, and

the low PM$_{2.5}$ in winter 2015-2016 is associated with anomalously high southerly flow and precipitation (Figure S4). The

low PM$_{2.5}$ in FWP in the winter 2014-2015 is associated with anomalously high wind speed, low RH, and low temperature,

while the high anomalies in the winter 2016-2017 are associated with anomalously low wind speed, high RH, and high

temperature (Figure S5).

### 4.    Conclusions

Observations of fine particulate matter (PM$_{2.5}$) pollution in China from the extensive CNEMC network established in 2013

show large 2013-2018 decreases in apparent response to emission controls. Here we used a stepwise multiple linear

regression (MLR) meteorological model to investigate the meteorological contributions to these 6-year trends.





The CNEMC observations show 34-49% decreases in $PM_{2.5}$ in the five megacity clusters targeted by the Chinese government's Clean Air Action to reduce anthropogenic emissions. Concurrent observations of $SO_2$, CO, and $NO_2$ are qualitatively consistent with these $PM_{2.5}$ decreases being driven by drastic cuts in emissions from coal combustion. At the same time, there is large interannual variability driven by meteorology particularly in winter when $PM_{2.5}$ is highest.

We used the stepwise MLR meteorological model to relate $PM_{2.5}$ anomalies across China to wind speed, precipitation,
relative humidity (RH), temperature, and meridional velocity at 850 hPa (V850) as potential predictors. The model accounts for ~50% of the variance in the deseasonalized detrended $PM_{2.5}$ data, including 41-65% for the five megacity clusters. Application to the $PM_{2.5}$ time series shows that meteorological variability contributed significantly to the 6-year trends across China and in the megacity clusters. Removing meteorological variability as given by the MLR model also reduces the uncertainty in the trend that can be attributed to emission controls. Thus the 2013-2018 $PM_{2.5}$ decrease for Beijing-Tianjin-
Hebei is $9.3 \pm 1.8$ μg m$^{-3}$ a$^{-1}$ in the original data and $8.0 \pm 1.1$ μg m$^{-3}$ a$^{-1}$ in the meteorology-corrected data. For the Sichuan Basin where the meteorological correction is particularly large, the $PM_{2.5}$ decrease is $6.7 \pm 1.3$ μg m$^{-3}$ a$^{-1}$ in the original data and $4.9 \pm 0.9$ μg m$^{-3}$ a$^{-1}$ in the meteorologically corrected data. The average 2013-2018 $PM_{2.5}$ decrease over our study domain is 5.2 μg m$^{-3}$ a$^{-1}$ in the original data (Figure 1 (right panel)), and is reduced by 12% to 4.6 μg m$^{-3}$ a$^{-1}$ in the meteorology-corrected data (Figure 5 (right panel)).

Observations for the 2015-2017 period indicate a flattening of the $PM_{2.5}$ trend in the Pearl River Delta and an increase in the Fenwei Plain. We find from the MLR model that these 3-year trends can be explained by meteorological variability (including particularly steep 2013-2015 decreases) rather than by relaxation of emission controls.

**Data availability.** All of the measurements, reanalysis data are openly available for download form the websites given in the
main text. The anthropogenic emission inventory is available from www.meicmodel.org, and for more information, please contact Qiang Zhang (qiangzhang@tsinghua.edu.cn).

**Competing interests.** The authors declare that they have no conflict of interest.

**Author contributions.** SXZ, DJJ, and HL designed the study. SXZ developed the model, performed the simulations and analyses. XW, LS, KL, YZZ, and TLZ helped with scientific interpretation and discussion. KG helped with pollutants data
processing. SXZ and DJJ wrote the manuscript and all authors provided input on the paper for revision before submission.



**Acknowledgments.** This work is a contribution from the Harvard-NUIST Joint Laboratory for Air Quality and Climate (JLAQC). Hong Liao is supported by the National Natural Science Foundation of China (91744311). This work was also supported by the China Scholarship Council, National Key R & D Program Pilot Projects (2016YFC0203304) and Natural Science Foundation of China (41830965 & 91744209). We thank the MEIC (Multi-resolution Emission Inventory for China) team in Tsinghua University for providing the MEIC emission inventory. We acknowledge Loretta J. Mickley (Harvard University) and Jonathan Moch (Harvard University) for helpful discussions.

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





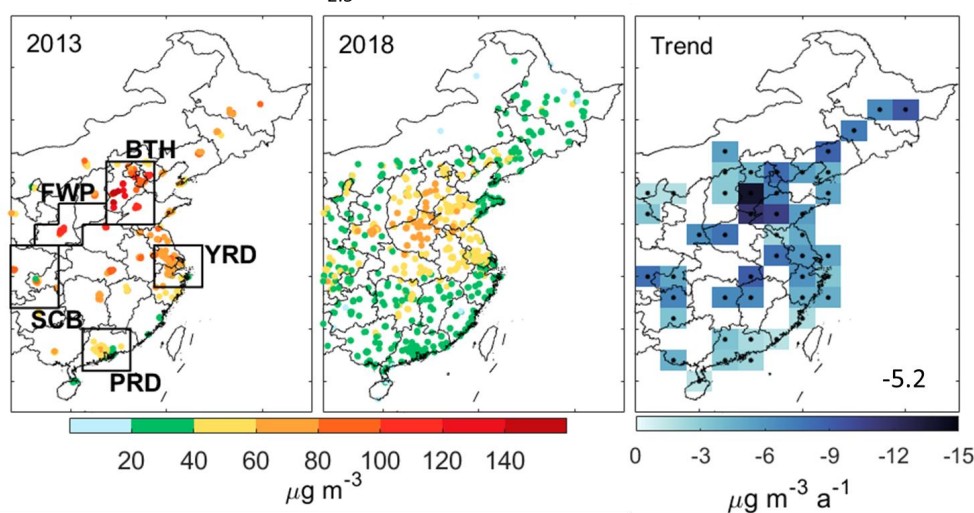

**Figure 1 Annual mean PM₂.₅ concentrations in China from the CNEMC network. Left and middle panels show values for 2013 and 2018 for sites with more than 70% data coverage for each year. The right panel shows the ordinary linear regression trends on a 2º×2.5º grid for sites with more than 70% data coverage for each of the five years from 2013 to 2018. The trends are based on the timeseries of 10-day mean anomalies as described in the text. Polygons in the left panel define the four target regions of the Clean Air Action (Beijing-Tianjin-Hebei (BTH; 35-41°N, 113.75-118.75°E), Yangtze River Delta (YRD; 29-33°N, 118.75-123°E), Pearl**

**River Delta (PRD; 21-25°N, 111.25-116.25°E), and Fenwei Plain (FWP; 33-35°N, 106.25-111.25°E & 35-37°N, 108.75-113.75°E)), to which we add Sichuan Basin (SCB; 27-33°N, 103.75-108.75°E). Number inset in the right panel is the trend in mean PM₂.₅ over the study region (21-41°N, 103.75-123°E). Dots in the right panel indicate grid squares with significant trends ($p < 0.05$).**





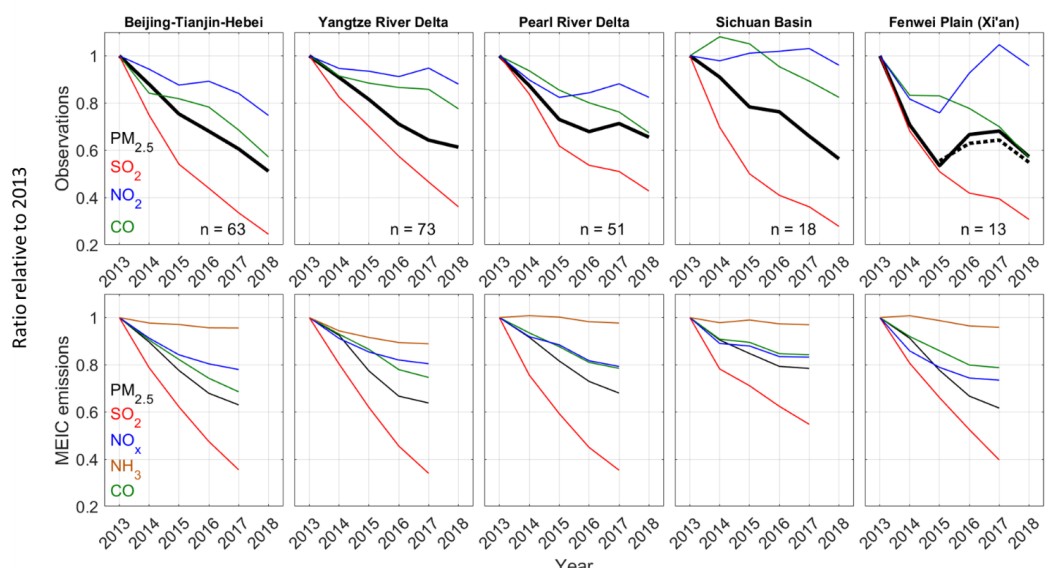

**Figure 2. Relative trends of 2013-2018 observed concentrations and 2013-2017 MEIC emission estimates for the five target regions of Figure 1. Values are annual means referenced to 2013. The observed concentrations are averaged over all sites in each region with at least 70% data coverage for each year. The number of sites for each region is indicated. Fenwei Plain trends are for Xi'an as other sites did not start operating until early 2015. Post-2015 relative PM$_{2.5}$ trends at these other sites are shown as the dashed line.**



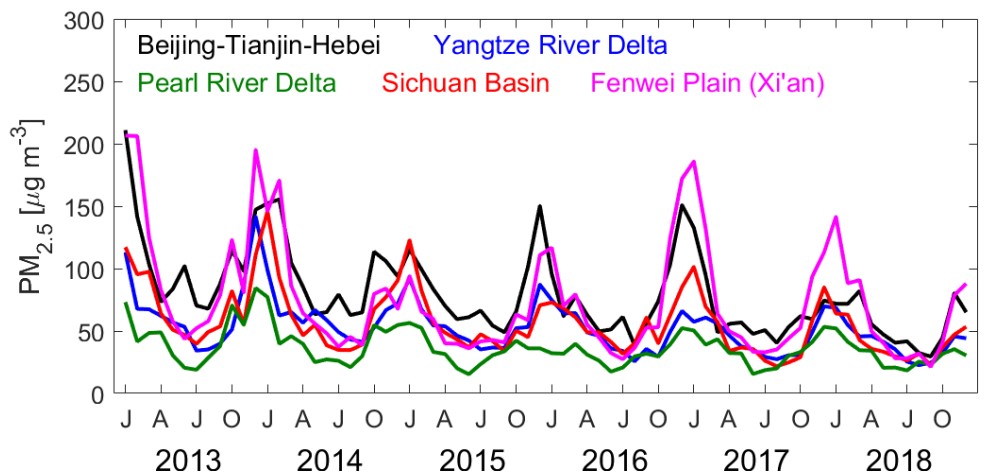

**Figure 3 2013-2018 time series of monthly mean PM₂.₅ concentrations over the five target regions. Values are averages from all sites in the region with over 70% data coverage for each year.**




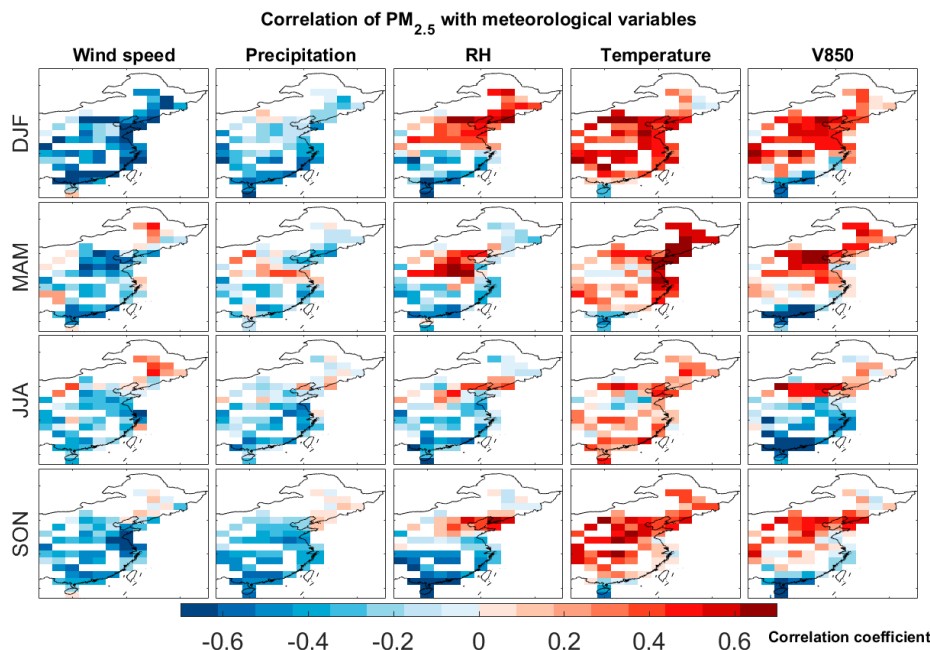


**Figure 4 Correlation coefficients of PM$_{2.5}$ concentration with the individual meteorological variables used in the MLR model: surface wind speed (m s$^{-1}$), precipitation (mm d$^{-1}$), relative humidity (RH; %), surface air temperature (°C), and 850hPa meridional wind velocity (m s$^{-1}$) for different seasons in China. The correlations are based on 10-day average observations on a 2°×2.5° grid.**






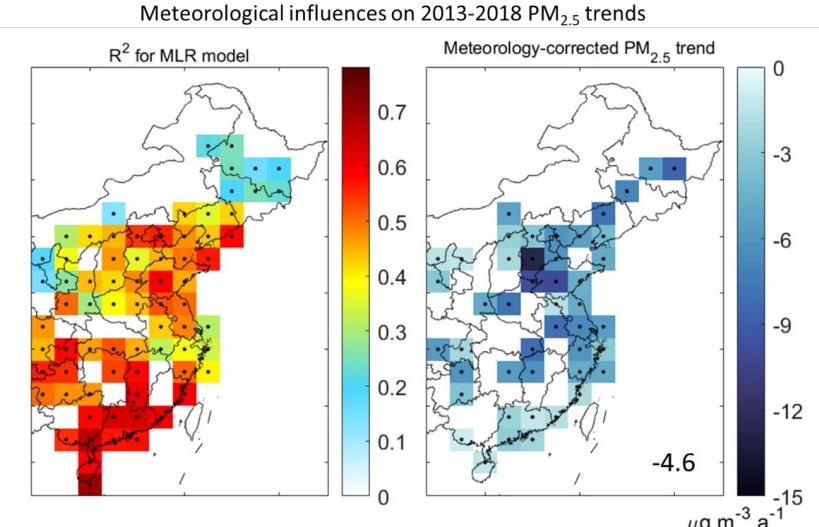

**Figure 5 Resolving meteorological influences on PM$_{2.5}$ 2013-2018 trends in China. The left panel shows the fraction of detrended and deseasonalized variance in 10-day PM$_{2.5}$ means explained by the stepwise multi linear regression (MLR) meteorological model. The right panel shows the meteorology-corrected trends, to be compared to the trends in the original data shown in Figure 1. Number inset in the right panel is the trend in mean PM$_{2.5}$ over the study region (same definition as in Figure 1). Dots indicate significant correlations ($p < 0.05$) in the left panel and significant trends ($p < 0.05$) in the right panel.**





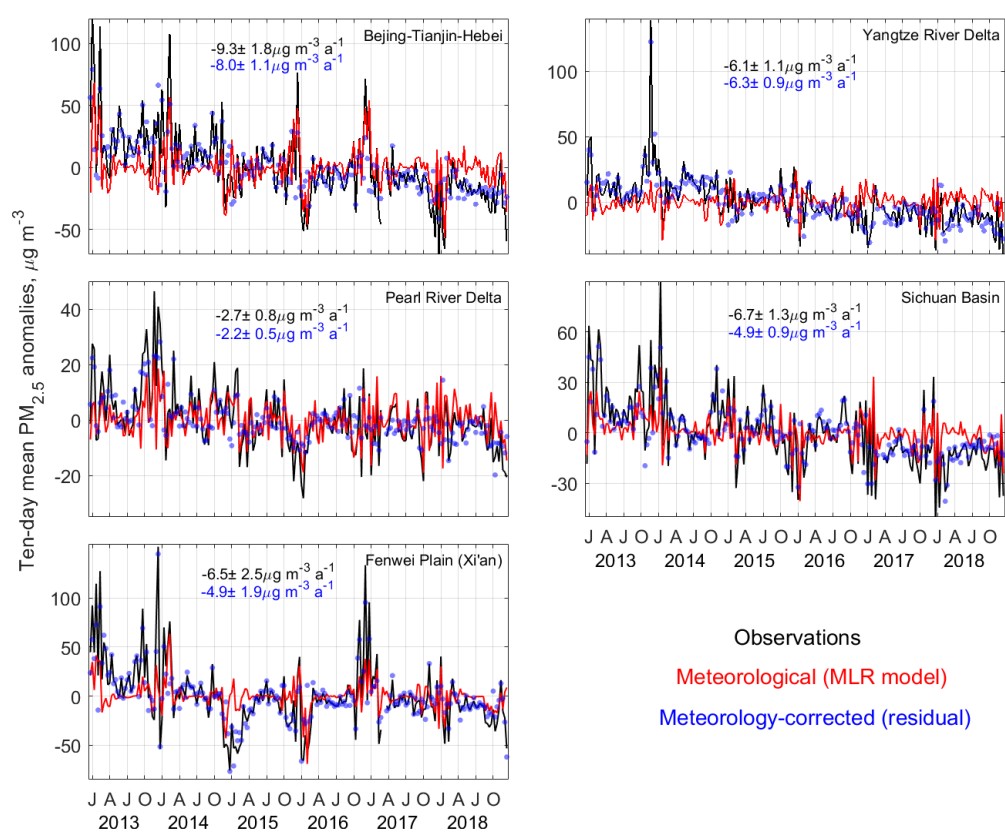

**Figure 6. Time series of 2013-2018 PM₂.₅ 10-day mean anomalies for the five target regions of Figure 1. The anomalies are relative**

**to the 2013-2018 means. The data are averaged over all measurement sites in each region with at least 70% of data coverage for**

**each year (same as for Figure 2). The meteorological contribution to the anomalies as diagnosed from the MLR model is shown in**

**red. The long-term trend in the meteorology-corrected residual in blue (equation (3)) is interpreted as driven by changes in**

**anthropogenic emissions. Values inset each panel are the ordinary linear regression trends with 95% confidence intervals obtained**

**by the bootstrap method.**
