# Peer review of "Fine particulate matter $(PM_{2.5})$ trends in China, 2013-2018: separating contributions from anthropogenic emissions and meteorology"

_Atmospheric Chemistry and Physics, 2019_

## Referee Comment (RC1) · Anonymous Referee #1 · 4 Jun 2019

General comments

This manuscript attempts to distinguish contributions from meteorology and emissions reduction to PM2.5 trends from 2013 to 2018 in five target regions in China. A multiple linear regression model (MLR) is developed using de-seasonalized (by taking 10-day average of hourly data) and detrended (by subtracting 50-day moving average of 10-day average from 10-day average) PM2.5 observations and corresponding five meteorological variables. The coefficients and intercepts obtained for each season and grid are applied to de-seasonalized but not detrended anomalies of meteorological variables (i.e., 50-day moving average minus 6-year average) to calculate PM2.5 anomalies attributable to meteorology. Consequently, residual anomalies are attributed to other factors, mainly changes in emissions. The attempt is valuable as the research

question, contribution from meteorology to the PM2.5 trend, is crucial to East Asian countries. Overall, the results with the MLR is acceptable. I would support publication of this manuscript with minor revision mostly asking clarification.

Specific comments

1) L25 'minor but significant': ambiguous expression. Can you add more explanation?

2) L26 'residual anthropogenic trends': anthropogenic emissions?

3) Section 2.3:

You may consider adding another variable for grid. For now, i represents both season and grid which made me difficult to follow at first.

Explicit description of $Y_{a,i}(t)$ is needed. It is not clear to me whether the anomaly is $Y_{a,i}(t)$ = 50-day moving average – 6-year average at the grid or $Y_{a,i}(t)$ = 10-day averge – (50-day moving average – 6-year average) at the grid.

4) Figure S2: How come PM2.5 anomalies are greater than de-seasonalized and de-trended PM2.5? It makes sense if $Y_{a,i}(t)$ is as the second definition as I mentioned above.

Technical corrections

L131 K. Li et al. (2019): Couldn't find this reference. Did you mean Yi et al. (2019)?

---

## Referee Comment (RC2) · Anonymous Referee #2 · 2 Jul 2019

In this study Zhai et al. use a combination ofPM2.5 observations and multiple linear regression modelling to analyse the trend in PM2.5 concentration across mainland China during 2013-2018 and to quantify the meteorological contribution to this trend. Overall the paper is well thought-out and written, and figures are well presented. The topic of the study is interesting and well within the scope of ACP. I recommend publication once the comments below (mostly regarding the processing of the data) have been addressed.

1. Abstract, L25: I suggest specifying that the contribution is "statistically" significant, otherwise the sentence reads a bit odd.

2. Abstract, L26: I think the statement "reduces the uncertainty on the emission-driven trends" needs more explanation in the abstract. Earlier in the abstract you refer to the

difficultly of trend attribution because of the meteorologically driven interannual variability in PM2.5 concentrations. However, it is not immediately clear what you mean by "uncertainty on the emission-driven trends". (It is worded more clearly in the conclusions section).

3. Introduction, L52: Please add an explanation to why the PM2.5 concentration is correlated to V850, particularly in the NCP.

4. Introduction, final paragraph: References to a few papers that have identified/quantified recent trends in PM2.5 concentrations across China seem to be missing from the introduction (Ma et al., 2019; Silver et al., 2018; Liang et al., 2016).

5. Section 2.1, L88: Can you give any example references here for these previous studies?

6. Section 2.1, L90: Why was 70% chosen? It seems quite low to me. Please add some justification. Did you do any sensitivity tests changing the threshold to a higher percentage?

7. Section 2.1, L90-91: As above, can you add some justification for the coarse grid chosen? Is this recommended by Tai et al. (2012)? Did you test any other grid resolutions?

8. Section 2.1: I see that you removed severe outliers from the observation dataset but what did you do about repeating consecutive values in the dataset (e.g. identified in Rohde and Muller (2015)) and day-to-day repeating sequences of values (e.g. identified by Silver et al., 2018)? If these were not removed, please at least acknowledge that data issues are likely remain in the dataset.

9. Section 2.1: If each year of data and each station were considered separately when applying the 70% data threshold, what does the introduction of more stations/data towards the end of the sampling time period (as more stations come online) do to the trends? I realise the data is averaged over large grid cells, but introduction of

many stations later in the time-series (that are not consistent) may impact the trends calculated. Please add some explanation. Did you attempt to calculate the trends based only on data from stations that were online in 2013?

10. Section 2.2 and 2.3: I am slightly confused by the explanation of the de-seasonalizing and detrending process. Perhaps the explanation could be reworded slightly? I have understood it as: the data is de-seasonalized and detrended by taking the 50-day moving average from the 10-day means; whereas the anomalies are de-seasonalized (but not detrended) by taking the 6-year mean 50-day moving average from the 10-day means; is this correct?

11. Section 3.1: I think this section is really nice and gives some good explanations (with references) for the drivers of the changes in pollutants and/or emissions. However, there is no comparison with previous studies that have calculated trends in PM2.5 concentrations over similar time periods (e.g. Ma et al., 2019; Silver et al., 2018; there may be others). Are the calculated trends consistent between studies, despite differences in the data or data processing? I understand that the trends are all calculated over slightly different time periods, but at least a qualitative comparison should be added to the text.

12. Conclusions: this section is a nice summary of the main points of the paper. However, it would make the results even clearer if the percentage difference from the original trend was quoted here again as in the abstract and it also might be worth explaining again here what is meant by meteorologically corrected data.
* * *

---

## Author Comment (AC1) · 20 Jul 2019

We thank the reviewers for their valuable comments. We have made efforts to improve the manuscript accordingly. This document is organized as follows: the referees' comments are in **Bold black**, our responses are in plain black text, and the revisions in the manuscript are shown in blue. The line numbers in this document refer to the updated manuscript.

Anonymous Referee #1

**General comments**

**This manuscript attempts to distinguish contributions from meteorology and emissions reduction to PM$_{2.5}$ trends from 2013 to 2018 in five target regions in China. A multiple linear regression model (MLR) is developed using de-seasonalized (by taking 10-day average of hourly data) and detrended (by subtracting 50-day moving average of 10-day average from 10-day average) PM$_{2.5}$ observations and corresponding five meteorological variables. The coefficients and intercepts obtained for each season and grid are applied to de-seasonalized but not detrended anomalies of meteorological variables (i.e., 50-day moving average minus 6-year average) to calculate PM$_{2.5}$ anomalies attributable to meteorology. Consequently, residual anomalies are attributed to other factors, mainly changes in emissions. The attempt is valuable as the research question, contribution from meteorology to the PM$_{2.5}$ trend, is crucial to East Asian countries. Overall, the results with the MLR is acceptable. I would support publication of this manuscript with minor revision mostly asking clarification.**

**Specific comments**

**1) L25 'minor but significant': ambiguous expression. Can you add more explanation?**

Thanks. We have rephrased this part to:

The meteorology-corrected PM$_{2.5}$ trends after removal of the MLR meteorological contribution can be viewed as driven by trends in anthropogenic emissions. The mean PM$_{2.5}$ decrease across China is -4.6 μg m$^{-3}$ a$^{-1}$ in the meteorology-corrected data, 12% weaker than in the original data. The trends in the meteorology-corrected data for the five megacity clusters are: …

**2) L26 'residual anthropogenic trends': anthropogenic emissions?**

We have rephrased this sentence to:

The trends in the meteorology-corrected data for the five megacity clusters are: …

**3) Section 2.3: You may consider adding another variable for grid. For now, *i* represents both season and grid which made me difficult to follow at first. Explicit description of Y$_{a,i}$(t) is needed. It is not clear to me whether the anomaly is Y$_{a,i}$(t) = 50-day moving average – 6-year average at the grid or Y$_{a,i}$(t) = 10-day averge – (50-day moving average – 6-year average) at the grid.**

Thanks for pointing this out.

$Y_{a,i}$ = 10-day average – 6-year average of 50-day moving average;

An explanation in brackets (Line 134) is added to explain the way to obtain the PM$_{2.5}$ anomaly $Y_{a,i}$: Consider now the PM$_{2.5}$ anomaly Y$_{a,i}$ for grid square and season *i* obtained by deseasonalizing but

not detrending the PM$_{2.5}$ data (by removing the 6-year means of the 50-day moving averages), in the same way as for the meteorological variables.

**4) Figure S2: How come PM$_{2.5}$ anomalies are greater than deseasonalized and detrended PM$_{2.5}$? It makes sense if Y$_{a,i}$(t) is as the second definition as I mentioned above.**

PM$_{2.5}$ anomalies ($Y_{a,i}$) can be greater than deseasonalized and detrended PM$_{2.5}$ ($Y_{d,i}$).

$Y_{a,i}$ = 10-day average – 6-year average of 50-day moving average;

$Y_{d,i}$ = 10-day average – 50-day moving average.

From above we can see that trends are not removed from $Y_{a,i}$ , and that both trends and seasonal variations are removed from $Y_{d,i}$. Therefore, the difference between PM$_{2.5}$ anomalies and deseasonalized and detrended PM$_{2.5}$ is that PM$_{2.5}$ anomalies contain trend information. This is clarified in the manuscript in Line130 as: "The anomalies calculated in this manner are deseasonalized but not detrended".

**Technical corrections**

**L131 K. Li et al. (2019): Couldn't find this reference. Did you mean Yi et al. (2019)?**

Thanks for pointing this out. We have added this reference in the reference section:

Li, K., Jacob, D. J., Liao, H., Shen, L., Zhang, Q., and Bates, K. H.: Anthropogenic drivers of 2013-2017 trends in summer surface ozone in China, Proceedings of the National Academy of Sciences, 116, 422-427, 2019.

---

## Author Comment (AC2) · 20 Jul 2019

We thank the reviewers for their valuable comments. We have made efforts to improve the manuscript accordingly. This document is organized as follows: the referees' comments are in **Bold black**, our responses are in plain black text, and the revisions in the manuscript are shown in blue. The line numbers in this document refer to the updated manuscript.

Anonymous Referee #2

**In this study Zhai et al. use a combination of PM$_{2.5}$ observations and multiple linear regression modelling to analyse the trend in PM$_{2.5}$ concentration across mainland China during 2013-2018 and to quantify the meteorological contribution to this trend. Overall the paper is well thought-out and written, and figures are well presented. The topic of the study is interesting and well within the scope of ACP. I recommend publication once the comments below (mostly regarding the processing of the data) have been addressed.**

**1. Abstract, L25: I suggest specifying that the contribution is "statistically" significant, otherwise the sentence reads a bit odd.**

Thanks. To make it clear, we have rephrased this part to:

The meteorology-corrected PM$_{2.5}$ trends after removal of the MLR meteorological contribution can be viewed as driven by trends in anthropogenic emissions. The mean PM$_{2.5}$ decrease across China is -4.6 ug m$^{-3}$ a$^{-1}$ in the meteorology-corrected data, 12% weaker than in the original data. The trends in the meteorology-corrected data for the five megacity clusters are: …

**2. Abstract, L26: I think the statement "reduces the uncertainty on the emission-driven trends" needs more explanation in the abstract. Earlier in the abstract you refer to the difficulty of trend attribution because of the meteorologically driven interannual variability in PM$_{2.5}$ concentrations. However, it is not immediately clear what you mean by "uncertainty on the emission-driven trends". (It is worded more clearly in the conclusions section).**

To make it clear, we deleted 'reduces the uncertainty on the emission-driven trends', and reworded this part in the abstract as:

The meteorology-corrected PM$_{2.5}$ trends after removal of the MLR meteorological contribution can be viewed as driven by trends in anthropogenic emissions. The mean PM$_{2.5}$ decrease across China is -4.6 μg m$^{-3}$ a$^{-1}$ in the meteorology-corrected data, 12% weaker than in the original data. The trends in the meteorology-corrected data for the five megacity clusters are: …

**3. Introduction, L52: Please add an explanation to why the PM$_{2.5}$ concentration is correlated to V850, particularly in the NCP.**

Explanation is added in lines 93-94: V850 in particular is a strong predictor of PM$_{2.5}$ wintertime pollution events in the North China Plain, because northerly winds (negative V850) ventilate the region with clean dry air (Cai et al., 2017; Pendergrass et al., 2019).

Added reference:

Cai, W., Li, K., Liao, H., Wang, H., and Wu, L.: Weather conditions conducive to Beijing severe haze more frequent under climate change, Nature Climate Change, 7, 257-263, 2017.

**4. Introduction, final paragraph: References to a few papers that have identified/quantified recent trends in PM₂.₅ concentrations across China seem to be missing from the introduction (Ma et al., 2019; Silver et al., 2018; Liang et al., 2016).**

Added. Thanks.

**5. Section 2.1, L88: Can you give any example references here for these previous studies?**

Example references (Wang et al., 2014; Cai et al., 2017; Shen et al., 2017; Leung et al., 2018; Song et al., 2019; Zou et al., 2017) are added.

**6. Section 2.1, L90: Why was 70% chosen? It seems quite low to me. Please add some justification. Did you do any sensitivity tests changing the threshold to a higher percentage?**

This threshold is aimed to include sites that have continuous observations since early 2013 (mainly sites locates in the 74 major cities). We have tried to use data from the 74 major cities and obtained identical trend results. I then improved the threshold to 80% and 90% and find that although the number of valid sites in each target region decreased a little bit, the pollutants trends have negligible differences compared with the trends when the '70% threshold' was used.

Added justification in Lines 96-97: We did sensitivity tests with data coverage thresholds changing from 70% to 90% and obtained similar pollutants trends. To make the most use of available data, 70% is chosen.

**7. Section 2.1, L90-91: As above, can you add some justification for the coarse grid chosen? Is this recommended by Tai et al. (2012)? Did you test any other grid resolutions?**

It is recommended by Tai et al. (2012) and Shen et al. (2017). I have tried to use 0.5°×0.625° grid resolution. However, finer resolution will result in too few valid grids (grids that have both PM₂.₅ and meteorology observations).

We have reworded the text in the manuscript as: For the MLR model, we further average all data on a 2°×2.5° grid to increase statistical robustness following Tai et al. (2012) and Shen et al. (2017).

**8. Section 2.1: I see that you removed severe outliers from the observation dataset but what did you do about repeating consecutive values in the dataset (e.g. identified in Rohde and Muller (2015)) and day-to-day repeating sequences of values (e.g. identified by Silver et al., 2018)? If these were not removed, please at least acknowledge that data issues are likely remain in the dataset.**

Reply: Thank you for pointing this out. I then checked the impacts of those consecutive repeats on this study. It turned out that these consecutive repeating values have negligible impacts on results in this study. Nevertheless, consecutive repeats identified by Rohde and Muller (2015) and Silver et al. (2018) are unlikely 'realistic' values, and are then removed throughout this study. We removed values from the hourly time series when there are >24 consecutive repeats. The threshold of '>24 consecutive repeats' were chosen by applying a series of thresholds from '>4' to '>24', and we find that different thresholds lead to negligible changes in this study results.

Changes in the manuscript (Line80-83):

At the end of Section 2.1: There are also occasional consecutive repeats of data that may be caused by faulty instruments or reporting (Rohde and Muller, 2015; Silver et al., 2018). Here we removed values

from the hourly time series when there are >24 consecutive repeats. This in whole removed 7.4%, 7.0%, 6.4%, and 6.7% of the PM$_{2.5}$, SO$_2$, NO$_2$, and CO data respectively.

Added reference:

Rohde, R. A., and Muller, R. A.: Air pollution in China: mapping of concentrations and sources, PloS one, 10, e0135749, 2015.

**9. Section 2.1: If each year of data and each station were considered separately when applying the 70% data threshold, what does the introduction of more stations/data towards the end of the sampling time period (as more stations come online) do to the trends? I realize the data is averaged over large grid cells, but introduction of many stations later in the time-series (that are not consistent) may impact the trends calculated. Please add some explanation. Did you attempt to calculate the trends based only on data from stations that were online in 2013?**

Thanks. This is a good point and we have already considered this in this study. For trend analysis, we only retained data from sites with at least 70% of data coverage for each year from 2013 to 2018. That is, the selected sites must have at least 70% data coverage for each of the 6 years from 2013 to 2018 simultaneously. In this way, we are using consistent sites throughout 2013-2018 for trend analysis. To make it clear, we have made modifications in Lines 96, 106 & 449: for each of the 6 years from 2013 to 2018.

**10. Section 2.2 and 2.3: I am slightly confused by the explanation of the deseasonalizing and detrending process. Perhaps the explanation could be reworded slightly? I have understood it as: the data is de-seasonalized and detrended by taking the 50-day moving average from the 10-day means; whereas the anomalies are deseasonalized (but not detrended) by taking the 6-year mean 50-day moving average from the 10-day means; is this correct?**

Yes, this is correct. Explanations in the manuscript are reworded as follows:

P4, Line 112-113: The deseasonalized and detrended time series are obtained by removing the 50-day moving averages from the 10-day mean time series.

P5, Line 129-130: We thus apply equation (1) to the meteorological anomalies $X_{a,i,k}$, obtained by removing the 6-year means of the 50-day moving averages from the 10-day mean time series.

**11. Section 3.1: I think this section is really nice and gives some good explanations (with references) for the drivers of the changes in pollutants and/or emissions. However, there is no comparison with previous studies that have calculated trends in PM$_{2.5}$ concentrations over similar time periods (e.g. Ma et al., 2019; Silver et al., 2018; there may be others). Are the calculated trends consistent between studies, despite differences in the data or data processing? I understand that the trends are all calculated over slightly different time periods, but at least a qualitative comparison should be added to the text.**

Thanks, the following line is added in P9, Line 168-169: Trends in China PM$_{2.5}$, SO$_2$, and NO$_2$ presented here are consistent with previous studies (Silver et al., 2018; Ma et al., 2019) that cover a shorter time period than 2013-2018.

**12. Conclusions: this section is a nice summary of the main points of the paper. However, it would make the results even clearer if the percentage difference from the original trend was**

**quoted here again as in the abstract and it also might be worth explaining again here what is meant by meteorologically corrected data.**

P9, Lines 241-245 are revised as:

We refer to the data series after removal of meteorological variability as the meteorology-corrected data. Thus the 2013-2018 $PM_{2.5}$ decrease for Beijing-Tianjin-Hebei is $-9.3 \pm 1.8$ µg m$^{-3}$ a$^{-1}$ in the original data and is 14% weaker in the meteorology-corrected data ($-8.0 \pm 1.1$ µg m$^{-3}$ a$^{-1}$). For the Sichuan Basin where the meteorological correction is particularly large, the $PM_{2.5}$ decrease is $-6.7 \pm 1.3$ µg m$^{-3}$ a$^{-1}$ in the original data and is reduced by 27% to $-4.9 \pm 0.9$ µg m$^{-3}$ a$^{-1}$ in the meteorology-corrected data.

Explanation of the meteorology-corrected data are also added in the Lines 24-26 and in Lines 137-138.

Additional changes:

1) References updated or newly added:

Cheng, J., Su, J., Cui, T., Li, X., Dong, X., Sun, F., Yang, Y., Tong, D., Zheng, Y., Li, Y., Li, J., Zhang, Q., and He, K.: Dominant role of emission reduction in $PM_{2.5}$ air quality improvement in Beijing during 2013-2017: a model-based decomposition analysis, Atmos. Chem. Phys., 19, 6125-6146, 10.5194/acp-19-6125-2019, 2019.

Song, S., Gao, M., Xu, W., Sun, Y., Worsnop, D. R., Jayne, J. T., Zhang, Y., Zhu, L., Li, M., Zhou, Z., Cheng, C., Lv, Y., Wang, Y., Peng, W., Xu, X., Lin, N., Wang, Y., Wang, S., Munger, J. W., Jacob, D. J., and McElroy, M. B.: Possible heterogeneous chemistry of hydroxymethanesulfonate (HMS) in northern China winter haze, Atmos. Chem. Phys., 19, 1357-1371, 10.5194/acp-19-1357-2019, 2019.

Zou, Y., Wang, Y., Zhang, Y., and Koo, J.-H.: Arctic sea ice, Eurasia snow, and extreme winter haze in China, Science Advances, 3, e1602751, 10.1126/sciadv.1602751, 2017.

2) Emission trends in Figure 2 for Pearl River Delta and Sichuan Basin are correctly reversed.

---

## Referee Report (RR1)

**General comments**

I would like to suggest adding a word 'detaching' or a synonymous in front of 'contributions' in the last part of title. The authors argue changes in PM2.5 trends when the meteorology is corrected throughout the manuscript rather than quantifying the amount of contribution from meteorology. Eventually readers could distinguish the contribution of meteorology at the end of the manuscript, the addition of a word would make it more straightforward based on how the manuscript was written.

Alternatively, expressions such as '12% weaker than in the original data' could be reworded as 'attributing additional 12% decrease to meteorology.' However, such expression was used frequently in the manuscript (e.g., L27-30, L199, L237) so the corrections would be somewhat broadscale and authors should pay great attention not to make confusion.

**Suggestions**

L40: complicated → biased

L68: Add a closing sentence describing the aim and the scope of this study.

L142 and Fig.1: colorbar is inappropriate to the explanation. Use finer scale.

L153: any discussion on increased CO in Sichuan in 2013-2015?

L167: northern China ← this refers which region defined in this study?

Section 3.2 and Fig.4: What about not showing -0.4 < r < 0.4 in the figure as they are not meaningful? Also, the explanation could be biased because correlation coefficients smaller than 0.4 were used when analyzing.

L188: other meteorological variables ← can you explain explicitly?

L212 and 215: it is difficult to see 2015-2017 flattening and increase from Fig.6. Looking together with Fig.2 helps.

Figure 2: why PM2.5 observation is thicker than others? Please explain.

**Technical corrections**

L20: CO → Carbon monoxide (CO)

L71: carbon monoxide (CO) → CO

---

## Author Response (AR2)

We thank the reviewers for their valuable comments. We have made efforts to improve the manuscript accordingly. This document is organized as follows: the referees' comments are in **Bold black**, our responses are in plain black text, and the revisions in the manuscript are shown in blue. The line numbers in this document refer to the updated manuscript.

**General comments**

**I would like to suggest adding a word 'detaching' or a synonymous in front of 'contributions' in the last part of title. The authors argue changes in PM$_{2.5}$ trends when the meteorology is corrected throughout the manuscript rather than quantifying the amount of contribution from meteorology. Eventually readers could distinguish the contribution of meteorology at the end of the manuscript, the addition of a word would make it more straightforward based on how the manuscript was written.**

Thanks. We revised the title to "Fine particulate matter (PM$_{2.5}$) trends in China, 2013-2018: separating contributions from anthropogenic emissions and meteorology".

**Alternatively, expressions such as '12% weaker than in the original data' could be reworded as 'attributing additional 12% decrease to meteorology.' However, such expression was used frequently in the manuscript (e.g., L27-30, L199, L237) so the corrections would be somewhat broadscale and authors should pay great attention not to make confusion.**

To make the expression clear, we added the expression following '12% weaker than in the original data' in the abstract as: "The mean PM$_{2.5}$ decrease across China is -4.6 μg m$^{-3}$ a$^{-1}$ in the meteorology-corrected data, 12% weaker than in the original data, meaning that 12% of the PM$_{2.5}$ decrease in the original data is attributable to meteorology".

In addition, we reworded the expression in Section 3.2: "The meteorology-corrected decreasing trend averaged across China is -4.6 μg m$^{-3}$ a$^{-1}$, 12% weaker than in the original data, meaning that 12% of the PM$_{2.5}$ decrease in the original data is attributable to meteorology. We elaborate below for the five target regions.".

**Suggestions**

**L40: complicated → biased.**

Revised.

**L68: Add a closing sentence describing the aim and the scope of this study.**

We rewrite the last sentence of the introduction to: "Our goal in this work is to quantify the response of PM$_{2.5}$ to these rapid emission changes by resolving the effect of meteorological variability, thus allowing improved assessment of the success of the Clean Air Action".

**L142 and Fig.1: colorbar is inappropriate to the explanation. Use finer scale.**

Figure 1 is replotted with finer scale in colorbar.

**L153: any discussion on increased CO in Sichuan in 2013-2015?**

We would rather not discuss this because it is peripheral to our general focus.

**L167: northern China: this refers which region defined in this study?**

We revised this to: "Residential heating emissions in winter also contribute to the seasonality in China north of about 33°N (covering BTH and FWP in this study; J. Liu et al., 2016b; Xiao et al., 2015)".

**Section 3.2 and Fig.4: What about not showing -0.4 < r < 0.4 in the figure as they are not meaningful? Also, the explanation could be biased because correlation coefficients smaller than 0.4 were used when analyzing.**

To address this concern, we have added dots in Fig.4 for grid squares with statistically significant correlations (p < 0.05). Meanwhile, a sentence is added in Line 194: "Correlation coefficients $r$ as low as 0.3 are statistically significant, and more so when consistent across a region".

**L188: other meteorological variables ← can you explain explicitly?**

We added "including wind speed, precipitation, and RH" following "other meteorological variables".

**L212 and 215: it is difficult to see 2015-2017 flattening and increase from Fig.6. Looking together with Fig.2 helps.**

A reference is added: "… 2015-2017 increase in the FWP (see Figure2) can be mostly attributed to …".

**Figure 2: why PM$_{2.5}$ observation is thicker than others? Please explain.**

We added after first sentence of caption: "The observed PM$_{2.5}$ trends are shown as thick lines".

**Technical corrections**

**L20: CO → Carbon monoxide (CO)**

Corrected.

**L71: carbon monoxide (CO) →CO**

Corrected.

[revised manuscript text omitted]